# Assessment of the Antibacterial Efficacy of Halicin against Pathogenic Bacteria

**DOI:** 10.3390/antibiotics10121480

**Published:** 2021-12-02

**Authors:** Rayan Y. Booq, Essam A. Tawfik, Haya A. Alfassam, Ahmed J. Alfahad, Essam J. Alyamani

**Affiliations:** 1National Center of Biotechnology, Life Science and Environment Research Institute, King Abdulaziz City for Science and Technology (KACST), P.O. Box 6086, Riyadh 11442, Saudi Arabia; rbooq@kacst.edu.sa (R.Y.B.); etawfik@kacst.edu.sa (E.A.T.); ajlfahad@kacst.edu.sa (A.J.A.); 2Center of Excellence for Biomedicine, Joint Centers of Excellence Program, King Abdulaziz City for Science and Technology (KACST), P.O. Box 6086, Riyadh 11442, Saudi Arabia; halfassam@kacst.edu.sa

**Keywords:** halicin, zone of inhibition, disc diffusion, minimum inhibitory concentration, multidrug-resistant bacteria, gram-positive, gram-negative bacterial strains

## Abstract

Artificial intelligence (AI) is a new technology that has been employed to screen and discover new drugs. Using AI, an anti-diabetic treatment (Halicin) was nominated and proven to have a unique antibacterial activity against several harmful bacterial strains, including multidrug-resistant bacteria. This study aims to explore the antibacterial effect of halicin and microbial susceptibility using the zone of inhibition and the minimum inhibition concentration (MIC) values while assessing the stability of stored halicin over a period of time with cost-effective and straightforward methods. Linear regression graphs were constructed, and the correlation coefficient was calculated. The new antibacterial agent was able to inhibit all tested gram-positive and gram-negative bacterial strains, but in different concentrations—including the *A. baumannii* multidrug-resistant (MDR) isolate. The MIC of halicin was found to be 16 μg/mL for *S. aureus* (ATCC BAA-977), 32 μg/mL for *E. coli* (ATCC 25922), 128 μg/mL for *A. baumannii* (ATCC BAA-747), and 256 μg/mL for MDR *A. baumannii*. Upon storage, the MICs were increased, suggesting instability of the drug after approximately a week of storage at 4 °C. MICs and zones of inhibition were found to be high (R = 0.90 to 0.98), suggesting that halicin has a promising antimicrobial activity and may be used as a wide-spectrum antibacterial drug. However, the drug’s pharmacokinetics have not been investigated, and further elucidation is needed.

## 1. Introduction

The rise of antibiotic-resistant bacteria is considered a crucial health threat globally [1]. The World Health Organization (WHO) has reported annual mortality of more than half a million people from antimicrobial-resistant infections [1]. Therefore, it is essential to develop novel approaches to overcome such a menace; one possible solution is to advance new antimicrobial drugs against multidrug-resistant (MDR) pathogens. This approach has encouraged the scientific community to engage and address the threat through various drug development methods. The process of discovering new applications for existing drugs [2] and using computational approaches [3], such as artificial intelligence (AI), have been utilized to find new antimicrobial agents. The latter approach generates and exploits algorithms that can link different genes and compositions by recognizing active reagents and observing their inhibition to microorganisms [4].

A team of scientists at MIT has recently reported using artificial intelligence to repurpose a new compound with antibacterial activities against broad spectrum bacteria, including MDR [5,6]. The team used a machine-learning algorithm AI from the ZINC15 database, an online collection of nearly 1.5 billion chemical compounds, to discover a previously developed anti-diabetic ‘Halicin’ (formerly known as SU-3327) with remarkable antibacterial activity against MDR bacteria [4]. Halicin is a c-Jun N-terminal protein kinase (JNK) inhibitor, which displayed unusual antibacterial mechanisms involving the disruption of the transmembrane electrochemical gradient and the upregulation of bacterial genes responsible for iron homeostasis, triggering the disturbance of the pH regulation across the bacterial cell membrane, which will halt bacterial growth. Hence, the bacteria may not be able to acquire resistance to this mechanism of action [7]. This discovery has opened the path to using AI in repurposing drugs that the drug authorities have approved to treat various infectious diseases, especially MDR bacteria.

As a result of a recent discovery, a couple of studies have evaluated the antibacterial activity of the drug. Unfortunately, the standard guidelines reference lacks the antibacterial activity of halicin against pathogenic bacteria, making it difficult to validate its antibacterial efficacy. Here, halicin MIC was validated and determined. The results of the antibacterial activity were correlated with the disc diffusion assay against gram-positive and gram-negative bacteria, including multidrug-resistant bacterium, to construct linear regression graphs of the halicin antibacterial drug.

## 2. Materials and Methods

### 2.1. Materials

Halicin was purchased from Carbosynth (Carbosynth Ltd., Compton, UK), while Mueller-Hinton broth was bought from Scharlab (Scharlab, S.L., Barcelona, Spain). Distilled water was generated through Milli Q (Millipore Corporation, Bedford, MA, USA) and had been used throughout this study. The measurement was made with CytationTM 3 Cell Imaging Multi-Mode Reader (BioTek Instruments, Winooski, VT, USA).

### 2.2. Preparation of Bacterial Suspensions

The American Type Culture Collection (ATCC) bacteria were used in this study and MDR clinical isolate. Muller-Hinton broth was used to revive the bacterial strains, *Staphylococcus aureus* (*S. aureus*; ATCC BAA-977), *Escherichia coli* (*E. coli*; ATCC 25922), and *Acinetobacter baumannii* (*A. baumannii*; ATCC BAA-747 as a control, and 3086 as an MDR clinical isolate from the Carbapenem-resistant patients in Saudi Arabia—resistant against beta-lactams, aminoglycosides, quinolones, tetracycline, and trimethoprim/sulfonamide). Bacterial suspensions were adjusted to a turbidity equivalent to that of a 0.5 McFarland Standard and incubated overnight under aerobic conditions at 37 °C. The media were prepared according to the manufacturer Scharlab’s recommendations [8].

### 2.3. Minimum Inhibitory Concentration (MIC) and Zone of Inhibition Assay

The MIC of halicin was determined by preparing a serial twofold dilution of the drug, ranging from 256 to 0.125 μg/mL, in Mueller–Hinton broth, and filtered by a 0.22 μm filter and added into 96-well plates containing bacterial suspensions. The total culture volume was 200 μL. A final bacterial inoculum of 1 × 10^6^ colony-forming-unit/mL (CFU/mL) was used, and incubated overnight at 37 °C with a continuous shaking speed of 120 RPM [8]. In addition, a similar serial dilution of halicin was prepared and stored at 4 °C for almost a week to assess the antibacterial efficacy upon reserving. The minimum concentration at which there was no bacterial growth (i.e., lack of turbidity) was considered the MIC; the plate was measured at 600 nm absorbance using CytationTM. This test was performed in independent duplicates.

The zone of inhibition assay was performed with discs impregnated with halicin solutions, at a twofold concentration range of 128 to 16 μg/mL against *S. aureus* (ATCC BAA-977) and *E. coli* (ATCC 25922), while the concentration range was 256 to 16 μg/mL against *A. baumannii* (ATCC BAA-747 and 3086 MDR isolate), according to the MIC results. A final concentration of 1 × 10^6^ CFU/mL inoculum was equally distributed on the Mueller-Hinton agar surface. The diameters of the clear areas of no growth were recorded in millimeters (mm). This test was performed in independent triplicates.

### 2.4. Statistical Analysis

The mean, standard deviation (SD), regression equation, correlation coefficient, and (R) coefficient of determination (R^2^) were calculated using Microsoft Excel 2016 software (Microsoft Corporation, Redmond, MA, USA). A t-test was used to compare the results’ variance, in which *p* < 0.05 was considered as statistically significant.

## 3. Results and Discussion

Drug repurposing is crucial in drug discovery, and it implicates comprehensive testing of drug activities pertinent to the disease occurrence. We aimed to demonstrate the antibacterial effect of halicin as an active ingredient against gram-positive and gram-negative, including MDR bacteria, by the zone of inhibition and MIC assays. These assays showed that the bacterial growth was suppressed or inhibited by the action of the tested substrate ‘halicin’. Simulating through the Clinical and Laboratory Standards Institute (CLSI) helped clarify the MIC with a sensitive inhibition zone diameter. Furthermore, these assays are characterized by their simplicity; they are fast and cost-effective compared to other microbiological tests. All of these factors are essential for the reproducibility of the data, mainly when testing different phenotypic bacteria.

Consistent with previous studies, halicin was able to inhibit the growth of gram-negative and gram-positive bacteria, including MDR strains [8]. The MICs of halicin were measured at 16 μg/mL for *S. aureus* (ATCC BAA-977), 32 μg/mL for *E. coli* (ATCC 25922), 128 μg/mL for *A. baumannii* (ATCC BAA-747), and 256 μg/mL for MDR *A. baumannii* (MDR 3086), as shown in Figure 1. These values were consistent with our previous study [8]. This variation in the MIC values is due to the different bacterial susceptibilities against the tested drug. Upon storage at 4 °C, the MICs were significantly increased to be 32 μg/mL for *S. aureus*, significantly increased to be 64 μg/mL for *E. coli*, insignificantly changed from 128 μg/mL for *A. baumannii*, and significantly increased to be ≥256 μg/mL for MDR *A. baumannii* (Figure 1). This result indicates that the efficacy of the halicin solution can be affected during its storage, as the *p* < 0.05 value presented. Hence, a stability evaluation is further required.

Due to the ease of its application and its low cost, the inhibition zone test is usually used to determine the sensitivity of the microbe that affects clinical decision-making. The result can be linked either to the ATCC strain samples, or compared with the CSLI or the European Committee on Antimicrobial Susceptibility Testing (EUCAST) reference platforms. On the other hand, the MIC test is relatively expensive since a higher amount of a drug is required to be measured. There is no reference evidence for halicin effectiveness against bacteria; hence, standard strains were used in this study to estimate the MIC and determine the region of inhibition on agar plates [9].

The inhibition zones demonstrated that halicin could inhibit all tested gram-positive and gram-negative bacterial strains, but with variable efficacy. The diameter of the zone of inhibition at a halicin concentration of 128 μg/mL was recorded as 29 mm for strains of *S. aureus* BAA-977, 20 mm for *E. coli* ATCC 25922, and 20 mm for *A. baumannii* ATCC BAA-747. However, the MDR *A. baumannii* was only sensitive at >128 μg/mL, with a diameter of 22 mm for a halicin concentration of 256 μg/mL. Table 1 summarizes the result of this assay. The correlation coefficient (R) for the zone of inhibition against all tested bacterial strains was determined, as shown in Table 1, to compare the diameters of the zones with dilution values of halicin, in which a value of ≥0.9 is considered a strong correlation.

The statistical coefficient of determination (R^2^) between halicin MIC and its bacterial inhibitory effect was also shown in Figure 2 [10]. This statistical model is based on the proportion of the total variation of outcomes explained by the model, and it showed the relation between halicin concentration and its bacterial inhibitory effect, which is considered strong (R^2^ ≥ 0.95) against *E. coli* ATCC 25922 and *A. baumannii* ATCC BAA-747, and fairly strong (R^2^ ≥ 0.8 and <0.9) against *S. aureus* ATCC BAA-977 and MDR *A. baumannii* MDR 3086.

Many factors might affect the reproducibility, susceptibility, and accuracy of the agar diffusion assay. Therefore, the interpretations consider all the variable factors [11]. The correlation of results from both assays can improve the accuracy and reproducibility of the data, and compensate for the breakpoint variations in the susceptibility testing that might occur due to handling error [12]. However, to compare between the halicin MIC and the antibacterial effect using zone of inhibition, Table 2 shows that *S. aureus* was sensitive to halicin at an inhibition diameter of ≥17 mm compared with the MIC (16 μg/mL), while *E. coli* was sensitive at a diameter of ≥14 mm at MIC 32 μg/mL. The *A. baumannii* ATCC strain was also sensitive at an inhibition diameter of ≥20 mm at MIC 128 μg/mL. In contrast, the MDR strain was resistant at the same concentration but sensitive at a higher concentration (265 μg/mL), which gave an inhibition diameter of >20 mm. These variable data suggest that the pharmacokinetics of the halicin drug may behave differently with sensitive and drug-resistant bacteria.

## 4. Conclusions

Halicin is effective against the tested gram-positive and gram-negative bacteria; *S. aureus’s* sensitivity was ≥16 mm, while both *A. baumannii* strains demonstrated a sensitivity of ≥20 mm. The *E. coli’s* sensitivity was ≥14 mm. Therefore, this study could be a reference for halicin antibacterial activity that will offer a guide when evaluating this drug’s pharmacological and toxicological windows. Nevertheless, the absence of a substantial analytical study and stability study is unexpected. Therefore, there is a need for a thorough investigation before the clinical use of halicin against MDR infections.

## Figures and Tables

**Figure 1 antibiotics-10-01480-f001:**
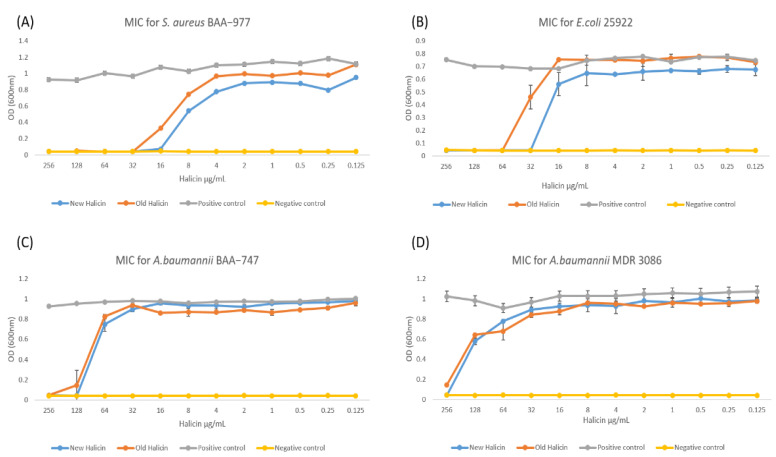
The line charts for minimum inhibitory concentration (MIC) values of fresh and old halicin against four types of bacteria showing the efficiency of halicin. The MICs of freshly prepared halicin were measured at 16 μg/mL for *S. aureus* ATCC BAA-977 (**A**), 32 μg/mL for *E. coli* ATCC 25922 (**B**), 128 μg/mL for *A. baumannii* ATCC BAA-747 (**C**), and 256 μg/mL for MDR *A. baumannii* MDR 3086 (**D**). In comparison, they doubled for the old-prepared halicin, which increased to be 32 μg/mL for *S. aureus*, 64 μg/mL for *E. coli*, 128 μg/mL for *A. baumannii*, and ≥256 μg/mL for MDR *A. baumannii.*

**Figure 2 antibiotics-10-01480-f002:**
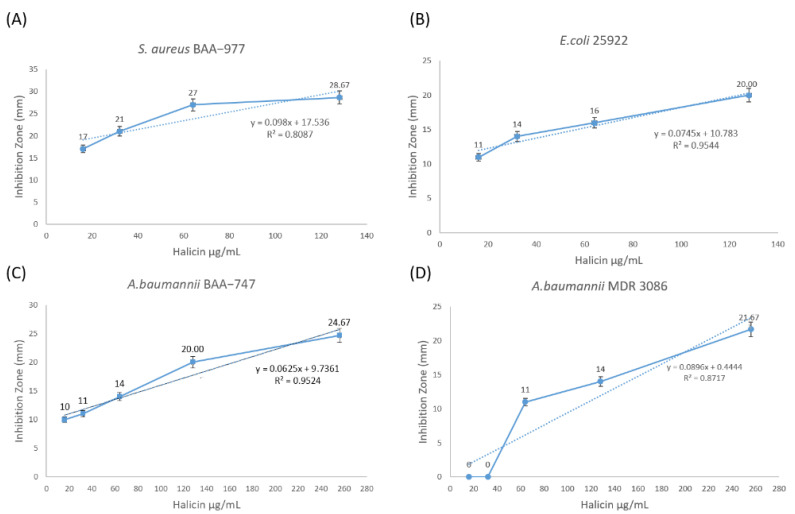
The line charts for the coefficient of determination (R^2^) values show the efficiency of halicin against four types of bacterial strains. The R^2^ of halicin was generally considered strong (R^2^ ≥ 0.9) against *E. coli* ATCC 25922 (**B**; R^2^ = 0.9544) and *A. baumannii* ATCC BAA-747 (**C**; R^2^ = 0.9524), and fairly strong (R^2^ ≥ 0.8 and <0.9) against *S. aureus* ATCC BAA-977 (**A**; R^2^= 0.8087) and *A. baumannii* MDR 3086 (**D**; R^2^ = 0.8717).

**Table 1 antibiotics-10-01480-t001:** The zone of inhibition diameters of halicin discs at a concentration range of 128 to 16 μg/mL against *S. aureus* ATCC BAA-977 and *E. coli* ATCC 25922, while the concentration range was 256 to 16 μg/mL against *A. baumannii* ATCC BAA-747 and 3086 MDR isolate. Halicin was effective against all the ATCC bacterial strains at all concentrations, while it was only effective at >128 μg/mL for the MDR *A. baumannii*. The data show a strong correlation coefficient between the MICs and their bacterial inhibitory activity. The results represent the mean (±SD) of *n* = 3.

Bacterial Strain	Halicin Concentration (μg/mL)	Zone of Inhibition (mm) Mean ± SD (*n* = 3)	Correlation Coefficient (R)
*S. aureus* ATCC BAA-977	16	17 ± 1	0.90
	32	21 ± 0	
	64	27 ± 1	
	128	29 ± 1	
*E. coli* ATCC 25922	16	11 ± 1	0.98
	32	14 ± 1	
	64	16 ± 1	
	128	20 ± 0	
*A. baumannii* ATCC BAA-747	16	10 ± 1	0.98
	32	11 ± 1	
	64	14 ± 0	
	128	20 ± 0	
	256	25 ± 1	
*A. baumannii* MDR 3086	16	0	0.93
	32	0	
	4	11 ± 1	
	128	14 ± 1	
	256	22 ± 1	

**Table 2 antibiotics-10-01480-t002:** The zone of inhibition diameters of halicin discs compared with MIC for *S. aureus* (ATCC BAA-977), *E. coli* (ATCC 25922), and *A. baumannii* (ATCC BAA-747 and 3086 MDR isolate).

Bacterial Strain	MIC (μg/mL)	Sensitive (mm)	Resistant (mm)
*S. aureus* ATCC BAA-977	16	≥17	≤15
*E. coli* ATCC 25922	32	≥14	≤11
*A. baumannii* ATCC BAA-747	128	≥20	≤14
*A. baumannii* MDR 3086	256	≥22	≤14

## Data Availability

The authors confirm that the data supporting the findings of this study are available within the article.

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
