# Peer review of "Assessment of the Antibacterial Efficacy of Halicin against Pathogenic Bacteria"

_antibiotics, 2021, doi:10.3390/antibiotics10121480_

Round 1

Reviewer 1 Report

The manuscript entitled “Assessment of the Antibacterial Efficacy of Halicin against Pathogenic Bacteria” by Rayan Y. Booq et al. described the investigation of antibacterial activity of halicin against tested bacterial strains of S. aureus (ATCC BAA-977), E. coli (ATCC 25922), A. baumannii (ATCC BAA-747), and A. baumannii multidrug-resistant (MDR) isolate. The manuscript may be of general interest to the researchers of this field, but the manuscript lacks some information that the author should consider and incorporate in the present form of the manuscript. Here are a few concerns that need to be addressed in the present form of the manuscript.

  1. “The new antibiotic” is not incorrect phrase. “The new antibacterial agent” is more suitable in this case.
  2. The phrase "gram-positive and gram-negative bacterial strains" should be added in keywords.
  3. There is no positive control for no one famous antibacterial drugs used in the determination of MICs and zones of inhibition. The antibacterial action of halicin against tested bacterial strains should be compared with reference drugs!
  4. Authors should add more information about clinical resistant strain of A. baumannii, for example, about the nature of origin, code of the strain, material, antibiogram—susceptibility profile, etc.
  5. Authors should once again carefully check the references in accordance with the examples reference style.

Author Response

The response is attached

Reviewer 2 Report

The work presents solid experimental evidence on the antimicrobial activity of halicin, both for ATCC bacteria strains and from clinical isolates.
Likewise, it presents, as a practical trial, the possibility of correlating the MIC with the growth inhibition halo for the bacterial strains tested.
As minor suggestions, before its potential publication, I tell the authors that:
1) in the caption of Figure 2 instead of "and quite strong (R2 ≥ 8)" it should say: "and quite strong (R20.8)"

2) The relationship between Figure 2 and Table 1 should be better explained.

3)On the other hand, it is very difficult to understand an R2 of 0.93 for A. Baumannii MDR 3086 from Table 2, with only two out of five data, different from zero. Also, one of them (3 ±6) without any statistical value.

Author Response

The response is attached

Round 2

Reviewer 1 Report

Dear colleagues,

many thanks for your respond to the suggestions from my side. I agree with your answers. 

 Wishing you all the best in future studies!